# The Urinary Metabolomic Fingerprint in Extremely Preterm Infants on Total Parenteral Nutrition vs. Enteral Feeds

**DOI:** 10.3390/metabo13090971

**Published:** 2023-08-24

**Authors:** Miguel Guardado, Martina Steurer, Cheryl Chapin, Ryan D. Hernandez, Philip L. Ballard, Dara Torgerson

**Affiliations:** 1Biological and Medical Informatics Graduate Program, School of Medicine, Mission Bay Campus, University of California, San Francisco, CA 94134, USA; 2Department of Epidemiology and Biostatistics, School of Medicine, Mission Bay Campus, University of California, San Francisco, CA 94158, USA; dara.torgerson@ucsf.edu; 3Department of Bioengineering and Therapeutic Sciences, School of Medicine, Mission Bay Campus, University of California, San Francisco, CA 94134, USA; ryan.hernandez@ucsf.edu; 4Department of Pediatrics, School of Medicine, Mission Bay & Parnassus Campuses, University of California, San Francisco, CA 94158, USA; martina.steurer@ucsf.edu (M.S.); cheryl.chapin@ucsf.edu (C.C.); phil.ballard@ucsf.edu (P.L.B.)

**Keywords:** prematurity, extreme premature birth, enteral feeding, total parenteral nutrition, untargeted metabolomic profiling

## Abstract

Total Parenteral Nutrition (TPN), which uses intravenous administration of nutrients, minerals and vitamins, is essential for sustaining premature infants until they transition to enteral feeds, but there is limited information on metabolomic differences between infants on TPN and enteral feeds. We performed untargeted global metabolomics on urine samples collected between 23–30 days of life from 314 infants born <29 weeks gestational age from the TOLSURF and PROP cohorts. Principal component analysis across all metabolites showed a separation of infants solely on TPN compared to infants who had transitioned to enteral feeds, indicating global metabolomic differences between infants based on feeding status. Among 913 metabolites that passed quality control filters, 609 varied in abundance between infants on TPN vs. enteral feeds at *p* < 0.05. Of these, 88% were in the direction of higher abundance in the urine of infants on enteral feeds. In a subset of infants in a longitudinal analysis, both concurrent and delayed changes in metabolite levels were observed with the initiation of enteral feeds. These infants had higher concentrations of essential amino acids, lipids, and vitamins, which are necessary for growth and development, suggesting the nutritional benefit of an enteral feeding regimen.

## 1. Introduction

Parenteral nutrition (PN) is an essential therapy for infants who are born extremely prematurely. These infants have minimal nutrient stores essential for growth and development plus an immature gastrointestinal system that limits nutrient uptake [1,2]. PN delivers all necessary nutrients—carbohydrates, proteins, lipids, vitamins, and minerals—directly into the bloodstream [3,4]. Almost all extremely premature infants start their postnatal life on total parental nutrition (TPN), and then gradually transitioned to enteral feeds when intestinal perfusion seems sufficient to absorb nutrients safely [5]. While TPN usage is a standard clinical practice for providing essential nutrition, the global impact on circulating metabolites is not well defined. 

TPN can also negatively affect extremely premature infants’ developing gut and metabolism. Potentially adverse effects include hyperglycemia, electrolyte abnormalities, acid–base disturbances, essential fatty-acid deficiency, and compromised gut integrity [6,7]. While providing nutrition through TPN is crucial for infants experiencing malnutrition, it is important to manage all aspects of nutrition carefully to avoid excessive intake that may result in negative consequences [8,9]. Understanding the influence that PN and especially TPN has on the metabolome of extremely preterm infants is useful for optimizing the timing and dosage of TPN [10]. 

Untargeted global metabolomics can be used to identify and quantify the levels of many hundreds of biochemicals in a biological fluid at a time [11,12]. An untargeted approach allows the systematic analysis of biofluid composition across a large number of metabolic pathways, eliminating the need to identify biochemical pathways in advance. Untargeted metabolomic approaches also allow for the identification of potential biomarkers from a large pool of candidates, distinguishing those that vary in response to a clinical intervention and capturing a wider range of physiological processes for biological inference [13,14]. Lastly, changes in the biochemical composition of the metabolome reflect the intersection of both environmental and genetic contributions, allowing a more comprehensive understanding of metabolomic changes that occur in response to different nutritional interventions [15,16].

We performed untargeted global metabolomics on urine samples from extremely premature infants on TPN vs. enteral feeds from two independent clinical cohorts. We hypothesized that there would be changes in the urinary metabolome as the infants were weaned off TPN to enteral feeding.

## 2. Materials and Methods

### 2.1. Study Description

Our study includes untargeted metabolomic profiling of 314 extremely premature infants from two cohorts: (a) The Trial of Late Surfactant for Prevention of Bronchopulmonary Dysplasia (TOLSURF, ClinicalTrials.gov, NCT01022580) and (b) The Prematurity and Respiratory Outcomes Program (PROP, NCT01435187). 

TOLSURF, was a blinded, randomized, sham-controlled trial performed between 2010 and 2013 at 25 U.S. centers to assess the effects of late surfactant treatment on respiratory outcomes. A total of 511 extremely low gestational age (<28 weeks) infants who required mechanical ventilation at 7–14 days were enrolled and received either surfactant treatment or sham installation every 1–3 days. A detailed description of the cohort, the trial design, infant characteristics, and effects of late surfactant treatment can be found elsewhere [17].

PROP was an observational study performed between 2010 and 2013 at 8 U.S. centers to collect clinical data and biospecimens for analysis of factors related to respiratory outcomes. A total of 835 infants <29 weeks were enrolled at 1–7 days with characteristics as previously described [18]. The research protocol for both studies was approved by the institutional review boards of the participating institutions and a parent of each infant provided written informed consent.

### 2.2. Metabolomic Profiling

We performed global metabolomic studies on single urine samples collected at day 23–30 from infants of both TOLSURF (*n* = 108 TPN feeds; *n* = 63 enteral feeds) and PROP (*n* = 80 TPN feeds, *n* = 63 enteral feeds) cohorts. TPN was initiated in all infants within the first two days of birth. In our study, the infants in the enteral feed group were required to have completely transitioned at least two days prior of the collection of their urine samples for metabolomic profiling. Detailed feeding data were available for PROP infants: most of those who were on full enteral feeds at week 4–5 (86.9%) received breast milk: 70.2% breast milk only, and 16.7% a mixture of breast milk and formula. Additionally, 28 infants in the TOLSURF study had longitudinal metabolomic profiling of their urine collected at 2–7 day intervals from 7–50 days after birth. All of these infants received breast milk as their enteral feed. Urine samples were obtained from infants by introducing cotton balls into their diapers. Following urination, a 10 mL syringe was employed to transfer the urine from the cotton balls into designated collection tubes. The samples were frozen at −70 °C and dispatched to Metabolon Inc (Morrisville, NC, USA) for untargeted metabolomic profiling, which was performed by Ultra High Performance Liquid Chromatography–Tandem Mass Spectroscopy (UHPLC-MS/MS). All methods used a Waters ACQUITY ultra-performance liquid chromatography (UPLC) and a Thermo Scientific Q-Exactive high resolution/accurate mass spectrometer interfaced with a heated electrospray ionization (HESI-II) source and Orbitrap mass analyzer operated at 35,000 mass resolution. Each reconstitution solvent contained a series of standards at fixed concentrations to ensure injection and chromatographic consistency. One aliquot was analyzed using acidic positive -ion conditions, chromatographically optimized for more hydrophilic compounds. In this method, the extract was gradient eluted from a C18 column (Waters UPLC BEH C18–2.1 × 100 mm, 1.7 µm) using water and methanol, containing 0.05% perfluoropentanoic acid (PFPA) and 0.1% formic acid. Another aliquot was also analyzed using acidic positive-ion conditions optimized for more hydrophobic compounds. In this method, the extract was gradient eluted from the same C18 column using methanol, acetonitrile, water, 0.05% PFPA and 0.01% formic acid and was operated at an overall higher organic content. Another aliquot was analyzed using basic negative-ion-optimized conditions using a separate dedicated C18 column. The basic extracts were gradient eluted from the column using methanol and water, with 6.5 mM ammonium bicarbonate at pH 8. The fourth aliquot was analyzed via negative ionization following elution from a HILIC column (Waters UPLC BEH Amide 2.1 × 150 mm, 1.7 µm) using a gradient consisting of water and acetonitrile with 10 mM ammonium formate, pH 10.8. The MS analysis alternated between MS and data-dependent MS^n^ scans using dynamic exclusion. The scan range varied slightly between methods but covered 70–1000 m/z.

Urine samples were normalized to osmolality and missing values were imputed to the minimum detectable level. For quality control, we only included metabolites in our study if they had been profiled in at least 70% of infants in each cohort separately (Appendix A). Data were normalized to the median value for each biochemical.

### 2.3. Global Metabolomic Analysis

Using the 913 metabolites in our study, we performed principal component analysis (PCA) [19] to visualize global differences in the urinary metabolome of infants on TPN vs. enteral feeds. Our metabolomic profiling was done in two batches for TOLSURF and PROP separately. For the PROP cohort, we included additional anchor samples from the TOLSURF cohort to merge the two batches into a joint metabolomic matrix, allowing us to compare metabolite abundances between the cohorts. We performed PCA on a matrix of 314 Infants × 913 Metabolites using Sklearn in Python 3.8. We then performed Unifold Manifold Approximation and Projection (UMAP) for dimension reduction using the raw metabolomic matrices in the UMAP package in Python 3.8. PCA and UMAP analyses were both implemented and performed in Python, and the source code is available on GitHub. A Mann–Whitney U-test was used to test for spatial separation between infants on TPN vs. enteral feeds for the first two PCs and UMAP coordinates in Python.

### 2.4. Metabolome Wide Association Analysis (MWAS)

We performed a metabolome-wide association study using logistic regression to identify biochemicals that vary in abundance by feeding mode (enteral vs. TPN), adjusting for potential confounders of clinical importance: birthweight, sex, corticosteroid treatment, and maternal self-identified race/ethnicity. Analyses were performed within each study separately, and results combined in a meta-analysis to account for study and experimental batch effects between the two cohorts (TOLSURF and PROP) (Appendix A). MWAS was implemented using a custom Python 3.8 script using the statsmodel package for logistic regression analysis. Meta-analyses using a weighted inverse variance were performed using the metafor package in R using summary statistics from MWAS analyses across TOLSURF and PROP. All source code is available on GitHub.

### 2.5. Metabolite Set Enrichment Analysis

We performed a metabolite set enrichment analysis (MSEA) to identify metabolic pathways enriched for metabolites that vary between infants on TPN vs. enteral feeds following Bonferroni correction for 913 tests (*p*-value < 5.47 × 10^−5^). Identified (named) metabolites from Metabolon were annotated to nine super-pathways and 92 sub-pathways. For each pathway, we tested for the enrichment of metabolites from our MWAS using a Fisher Exact Test, adjusting for multiple comparisons by inferring the false discovery rate (FDR) using the q value package in R.

### 2.6. Time Course Comparison Analysis

For 28 infants in the TOLSURF study, we conducted longitudinal metabolomic profiling of their urine between 7 and 50 days after birth. For each infant, the age at full enteral feed was recorded with the weaning process having begun approximately a week before. We compared two-time points, one before and one after stopping TPN, for each infant to quantitate the impact of the transition to enteral feeding [20]. The first time point was at least five days before TPN was stopped (TP1), and the second was 0–2 days after starting a full enteral feed (TP2). Five infants were excluded due to not having timepoints in the inclusion criteria, which left 23 infants for the study. We used TP2/TP1 to generate a fold change in the effect for each metabolite, and the abundance of each metabolite was compared between timepoints using a paired Student’s *t*-test (Appendix A).

## 3. Results

Overall, we identified 1437 biochemicals, of which 913 passed our quality control filters and were detected in at least 70% of all infant samples (Appendix A). Of the 913, 689 were named metabolites (by Metabolon) as of July 2022. Clinical characteristics of infants included in our study are shown in Table 1, including maternal self-reported racial/ethnic groups that were balanced between the comparison groups. Infants on hydrocortisone treatment were more likely to be on TPN at the time of metabolomic profiling than those on enteral feeds (*p* = 0.0029); thus, we adjusted for hydrocortisone treatment in addition to birthweight, sex, and maternal self-identified race/ethnicity.

Global analyses over all metabolites using dimension reduction showed both a qualitative separation of infants by feeding status (Figure 1a), and a statistically significant separation for both PC1 (*p* = 1.51 × 10^−15^) and PC2 (*p* = 4.36 × 10^−9^) (Figure 1b). UMAP showed a similar separation of infants by feeding status across both latent dimensions (Appendix A). These results highlight the global metabolomic shift between infants who are on TPN vs. enteral feeds (primarily breast milk).

In our analyses of individual metabolites, we found that 609 of the 913 (67%) were associated with infant feeding status at *p* < 0.05 (Figure 2), consistent with the separation of infants by feeding status in our global metabolomic comparisons. Of the 609 metabolites that varied by feeding status, 309 (51%) were still significant when applying a conservative Bonferroni threshold to adjust for multiple comparisons across 913 tests (*p* < 5.5 × 10^−5^). Among metabolites that varied by feeding status with *p* < 0.05, the majority (88%) were enriched in infants on enteral feeds compared to TPN (Figure 2) (Appendix A).

The top 20 most significant associations with mode of feeding are shown in Table 2. Dexpanthenol (provitamin B5) and gluconate, which belong to the xenobiotic superpathway, had the strongest signals of association with feeding status and were enriched in infants on TPN. Both chemicals are components of PN as vitamins (dexpanthenol) and mineral (calcium gluconate) supplements, and their concentrations serve as positive controls for the level of TPN administration. We additionally identified pantothenate, a natural form of vitamin B5, as a top hit in our MWAS study and found higher levels for infants on enteral feeds. Eight of the top 20 hits were cofactors and vitamins, and levels were higher on enteral feeds for 6 of these chemicals. Other significant metabolites were members of lipid, amino acid, carbohydrate, and energy superpathways.

In the amino acid superpathway, 12 of 20 amino acids were significantly associated with infant feeding status, with concentrations of 11 being higher in infants on enteral feeds (avg FC = 1.59) (Figure 3a). Four of these were essential amino acids (not synthesized in vivo): lysine, leucine, threonine, and tryptophan. The only amino acid with significantly higher levels in the urine of infants on TPN feeds was cysteine (8.2-fold), a non-essential sulfur-containing amino acid important for protein synthesis and redox status [21].

Additionally, we identified differences in lipid metabolites involved in acylcarnitine pathways, which play a crucial role in energy metabolism and may contribute to necrotizing enterocolitis (NEC) [22]. We examined the impact of feeding status on acylcarnitines using the metabolites profiled by Sylvester et al. in their analysis assessing the impact of acylcarnitine on NEC [22]. C5 was profiled across three metabolites (isovalerylcarnitine, 2-methlbututyrylcarnitine, and pivalerylcarnitine), and two were profiled for C4: isobutyrylcarnitine and butyrylcarnitine (Figure 3b). Of the 21 metabolites used in their analysis, 14 were profiled in ours. We found 7 acylcarnitines significant in our analysis, with three of them belonging to the C5 subclass: pivaloylcarnitine, 2-methylbutryrlcarnitine, and isovalerylcarnitine.

Using super- and subpathway annotations provided by Metabolon, we performed pathway enrichment analysis to identify metabolomic pathways enriched for metabolites that varied by feeding status. We found two superpathways (energy and nucleotide) and eight subpathways significantly enriched in infants on enteral feeds (Figure 4). Of these eight, three were acylcarnitines: long-chain saturated, dicarboxylate, and dihydroxy fatty acids. Additionally, we found the ATP subpathway enriched as a component of the energy superpathway.

Finally, we examined the kinetics of changes in metabolite levels with weaning off TPN, which occurs over approximately 1 week. For 28 infants in the TOLSURF study, we conducted longitudinal metabolomic profiling of their urine (7–50 days after birth) (Appendix A). We first qualitatively determined the time course of change with the transition to enteral feeding for the top metabolites in each of eight superpathways identified in our MWAS analysis and compared the temporal pattern with that of dexpanthenol as a marker for an exogenous chemical in TPN. Each of these metabolites, except for dexpanthenol, was enriched in infants on enteral feeds. For the infant shown in Figure 5, dexpanthenol abundance increased over the first 20 days and then decreased dramatically (74-fold) as TPN feeds decreased over 3 days and full milk feeds began on day 23. Conversely, there were temporally related increases in concentrations of N-acetyl asparagine (3.3-fold, amino acid superpathway) and 3-methyladipate (2.2-fold increase, lipid superpathway) over the same three-day transition to enteral feeding (Figure 5a). For this infant, 4 metabolites demonstrated a rapid increase in concentration concurrent with the decrease in dexpanthenol, 3 metabolites had a delayed increase, and 1 metabolite (prolylglycine) demonstrated a steady increase in concentration over time independent of the transition in feeding status (Figure 5b–d).

We used two timepoint values, one before and one after stopping TPN, to quantitate the impact of the change in feeding status to complement our single timepoint/cross-sectional analysis. We found 461 metabolites across both approaches with a change in concentration in the same direction by each analysis (*p* < 0.05). Using linear regression, there was a strong correlation between the fold changes of metabolites (R^2^ = 0.37; *p* = 2.53 × 10^−83^) and *p* values (R^2^ = 0.40, *p* = 9.02 × 10^−93^) (Appendix A) comparing cross-sectional and longitudinal results.

## 4. Discussion

Our study used untargeted metabolomics to identify urinary metabolites that change in response to TPN vs. enteral feeds. Our data indicated a global and dramatic shift of multiple metabolomic pathways when comparing infants on TPN versus enteral feeds. We identified 609 out of 913 urinary chemicals with altered concentrations, and of these metabolites, 88% had higher concentrations for infants on enteral feeds. They were enriched across all nine metabolomic superpathways, indicating the wide scope of responses and that levels of many biochemicals changed concurrently with the gradual reduction of the TPN dose.

Prior studies with fewer infants assessed metabolomic differences by modes of feeding, which allowed comparisons with our results. A study by Esturau-Escofet identified eight metabolites in the urine of 34 preterm infants that varied by enteral vs. parenteral nutrition [23]. Of the eight, we observed similar differences in five metabolites in the same direction of being enriched in PN vs. enteral feeds (gluconate, N-acetyltyrosine, succinate, citrate, and lactose). However, this study included infants of a predominantly higher gestational age: 26–36 weeks, with only three infants born extremely premature at <29 weeks. Nilsson et al. examined levels of 31 blood biochemicals over time in extremely premature infants [24]. Of the eight amino acids also detected in our study (urine), only tyrosine showed a similar pattern of increased abundance on enteral feeds. However, an additional eight biochemicals were in agreement and were more abundant in infants on enteral feeds in both studies, which raised the question of different biological fluids (3-hydroxyisobutyrate, creatinine, glycerol, leucine, lysine, proline, threonine, and uridine). Finally, a study by Falaina et al. observed global shifts in the urinary metabolome of term infants exclusively fed breast milk vs. formula across the first 3 months of life [15], similar to the dramatic shift we observe between PN vs. enteral feeds. Thus, although prior studies varied substantially from our own, a number of similarities were observed in individual metabolites and observations of the global metabolic impact of feeding status. Highlights of our study included two cohorts of extremely premature infants with diverse maternal racial/ethnic backgrounds recruited from 35 different medical centers across the U.S. From a methodological perspective, our untargeted approach identified close to 1000 different urinary chemicals for univariate and longitudinal comparisons, allowing us to identify individual metabolites that change by type of nutrition while also adjusting for confounding factors.

This study illustrated the benefit of global metabolomic approaches in identifying and quantifying systemic effects of environmental factors such as diet, medications, and contaminants. The change in the human metabolome as infants transition to enteral feeds demonstrates the wide scope of both direct and indirect effects of a change in diet. The large percentage of chemicals that change in both detection and concentration with initiation of enteral feeds was not unexpected given the limited and defined composition of PN compared to both formulas, which contained bovine skim milk and whey concentrate (with any bound chemicals), even more so in breast milk, which contains chemicals derived from maternal diet, medications, and exposures. The change in metabolite concentration secondary to altered intake was illustrated by amino acid metabolism, where 11 of the 12 amino acids with altered levels were enriched for infants fed on milk; the only metabolite found enriched in infants on TPN feeds was cysteine [25].

Two lipid metabolites affected by feeding status in our study, C5 and C12 acylcarnitine, were associated with the risk of developing NEC, which is an inflammatory condition of the immature gastrointestinal mucosa in premature infants and often related to enteral feeds. Extremely premature infants often have an insufficient supply of carnitine due to their immature liver and kidneys, which can lead to toxic fatty acid build-ups. Recent work has shown that TPN increases the risk of developing NEC, suggesting a potential role in regulating fatty acid metabolism [22]. Interestingly, two metabolites were found in consensus directions with a developing risk for NEC and TPN usage. C5 was enriched in infants on TPN feed (OR_meta_ = 2.00), with higher levels associated with an increased risk of developing NEC (OR_NEC_ = 1.75). C12 was enriched in infants on enteral feeds (OR_meta_ = 0.73) with higher levels found to be protective against developing NEC (OR_NEC_ = 0.74). This finding highlighted the potential effect that TPN duration may have on levels of C5, which could contribute to the risk of developing NEC. However, it is unclear whether increased C5 with TPN contributes to the development of NEC or is a general marker for sicker premature infants. Further work will be needed to look at the relationship between acylcarnitine levels and NEC and to establish the optimal dose for use in PN [26].

We reported a preliminary survey of the time course of metabolite concentrations related to the weaning off TPN and initiating enteral feeds, including examples of both rapid and delayed changes in concentrations of biochemicals. The different temporal patterns likely reflected the result of a direct change in intake amount (e.g., isocitric lactone, rapid response) versus changes in concentration resulting indirectly from altered levels of a biochemical in the same pathway (e.g., lyxonate, delayed response). Additionally, delayed responses in circulating metabolites may have arisen from an altered composition of the gut microbiome secondary to the intake of milk, an area for future study.

The dramatic changes observed in the metabolome by feeding mode could have implications for newborn blood screening (NBS), which detects severe but treatable rare disorders to help infants avoid adverse outcomes [27]. NBS screening has evolved over the years but still involves mass spectrometry on blood samples to identify inborn errors of metabolism or other inherited disorders [27,28]. Previously, strong associations between measured NBS metabolites and persistent pulmonary hypertension, hyperbilirubinemia, and NEC have been reported [26,27,28,29]. Recently, Oltman et al. described a metabolomic vulnerability profile that combines clinical characteristics and selected metabolites from NBS to predict the occurrence of common complications of preterm birth as well as overall morbidity and mortality [30]. Their model included 16 metabolites that were measured as part of NBS: 7 amino acids and 9 acylcarnitines. In our study we found that concentrations of 8 of these metabolites —glycine, leucine/isoleucine, proline, tyrosine, and the acylcarnitines C3, C4, C5, and C12—were influenced by feeding status. Because NBS screening in premature infants can occur between 1 and 7 days after birth, these infants may be receiving full TPN or enteral feeding or a mixture of the two, which will influence metabolite levels. Thus, the predictive value of vulnerability profiles using NBS data will be influenced by feeding status, and ideally this information is collected and used for adjustments of predictive profiling.

There were some limitations to our study. The TOLSURF and PROP clinical studies were conducted between 2010 and 2013. Thus, the urine samples were stored (at −70 °C) for up to 10 years, and it is possible that oxidation or degradation of some biochemicals occurred. Even though we used two clinical cohorts spanning multiple medical centers, larger and more inclusive cohorts should replicate this analysis and highlight population-specific effects of infant metabolism. Additionally, for all the metabolites associated with our analysis that may have clinical importance, targeted analyses should be performed to validate the effect of feeding status on metabolite abundances. Another limitation of this analysis was the inability, due to lack of clinical data, to compare the effect of breast milk vs. formula milk on the urinary metabolome. Future studies can address this interesting topic [31]. It’s also important to note that infants on TPN might be sicker than infants who transitioned to enteral feeds, which might have affected metabolism. Although we adjusted for BPD, future study to determine a causal relationship is warranted. Furthermore, our study did not provide data regarding the longitudinal weight gain of infants exposed to the two different nutritional sources. This limited our ability to evaluate the metabolomic effects of infant weight gain. Subsequent research should aim to investigate this aspect. Lastly, our study had a limitation regarding the exclusion of rare biochemicals. Our method of imputation resulted in left-censored data, whereby undetected metabolites were assigned a value equal to the minimum level of detection. However, it is important to note that we identified an additional 204 metabolites of significance below our imputation quality threshold. These biochemicals were profiled in less than 70% of the population (Appendix A). It is crucial for future research to address this issue by incorporating rare biochemicals into metabolomic association studies.

## 5. Conclusions

In conclusion, we observed a large difference in the urinary metabolome of extremely premature infants on TPN versus enteral feeds, primarily in increased levels of biochemicals in infants on enteral feeds. The results of this study support the common clinical practice of transitioning neonates from TPN to enteral feeds as soon as they are clinically able. This study demonstrated how metabolomics can enhance precision medicine, showcasing a molecular investigation in harmony with established clinical understanding.

## Figures and Tables

**Figure 1 metabolites-13-00971-f001:**
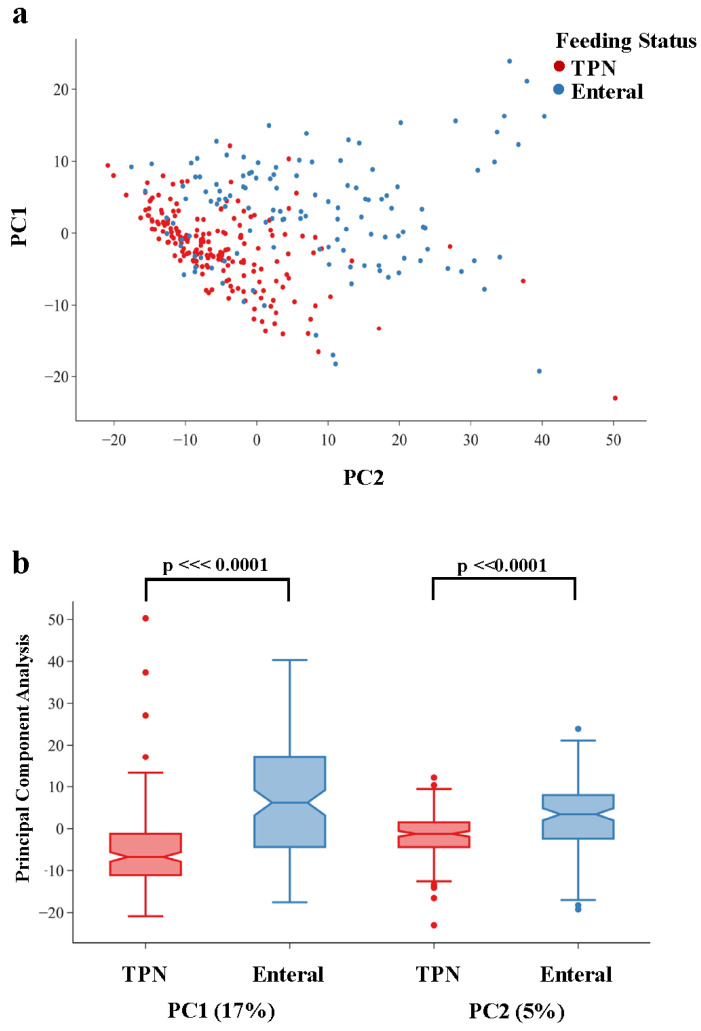
Results of Principal Component Analysis (PCA) of quantitative levels of 913 biochemicals detected in the urine of infants from the TOLSURF and PROP cohorts between 23 and 30 days of life. Infants who were on TPN at urine collection are shown in red and those on enteral feeds at urine collection are shown in blue. (**a**) X–Y plot of the first two Principal Components (PC1 and PC2) for each infant, (**b**) Boxplot of PC1 and PC2 by infant feeding status showing significant differences between TPN vs. enteral feeds for both PC1 (*p* = 1.51 × 10^−16^) and PC2 (*p* = 4.36 × 10^−9^).

**Figure 2 metabolites-13-00971-f002:**
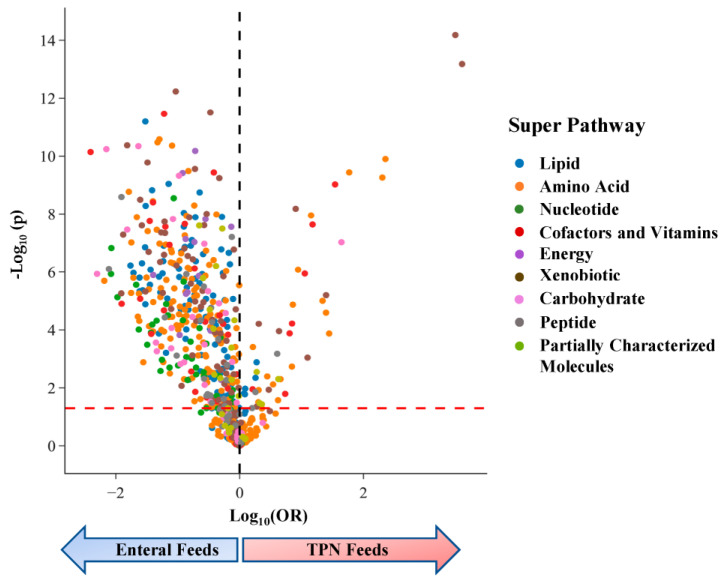
Volcano plot from MWAS using logistic regression to compare urinary profiles of infants on enteral feeds vs. TPN. The dashed red line represents *p* = 0.05; metabolites with an odds ratio (OR) > 0 are higher in infants on TPN, and metabolites with OR < 0 are higher in infants on enteral feeds (blue arrow). Metabolites are significantly enriched in infants who transitioned to enteral feeds for all super-pathways.

**Figure 3 metabolites-13-00971-f003:**
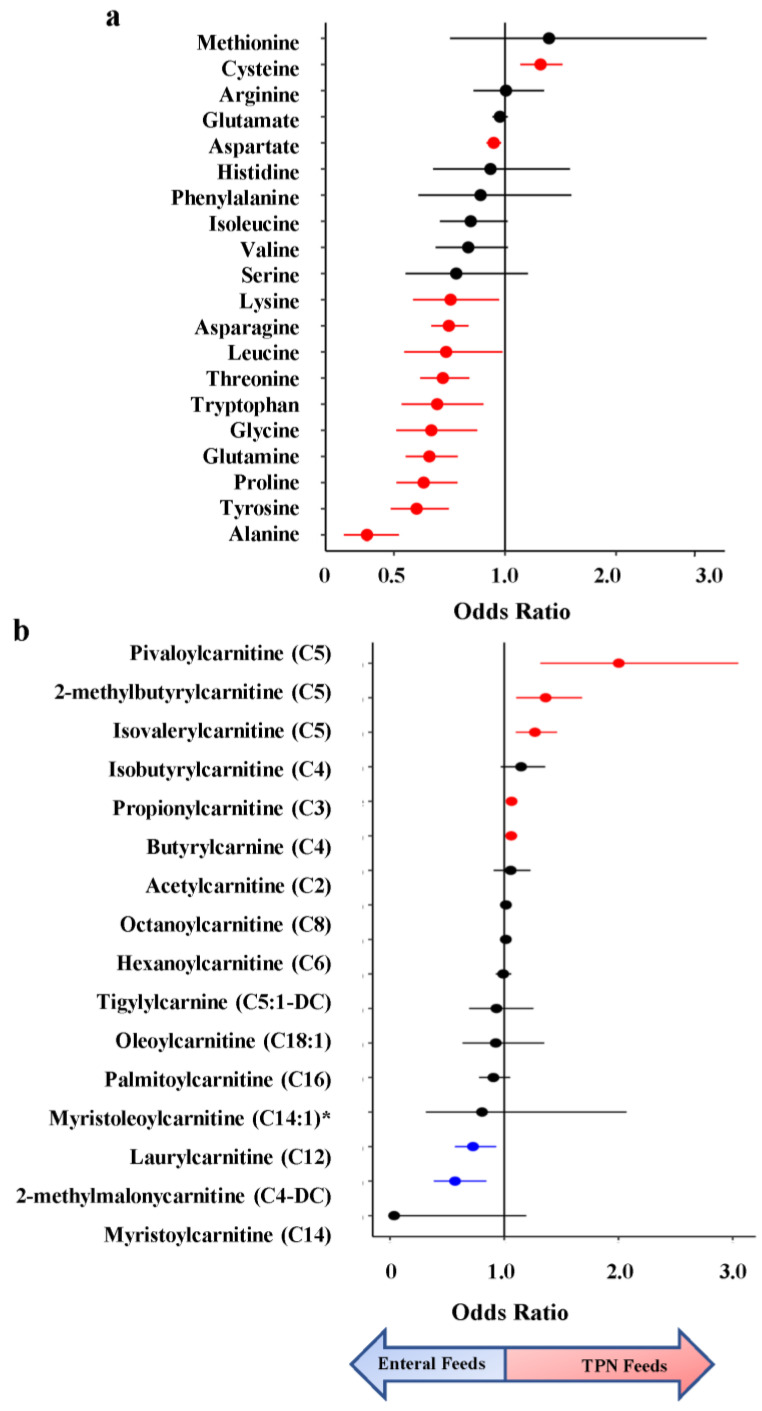
Odds Ratio (OR) of amino acids and acylcarnitines from metabolomic association analysis of TPN vs. enteral feeding. (**a**) OR estimates for amino acids, with essential amino acids colored in red. (**b**) OR estimates for acylcarnitines. Metabolites significantly enriched in enteral feeds are colored in blue and those enriched in TPN feeds are colored in red. * Indicates a compound that has not been confirmed based on a standard, but Metabolon is confident in its identity.

**Figure 4 metabolites-13-00971-f004:**
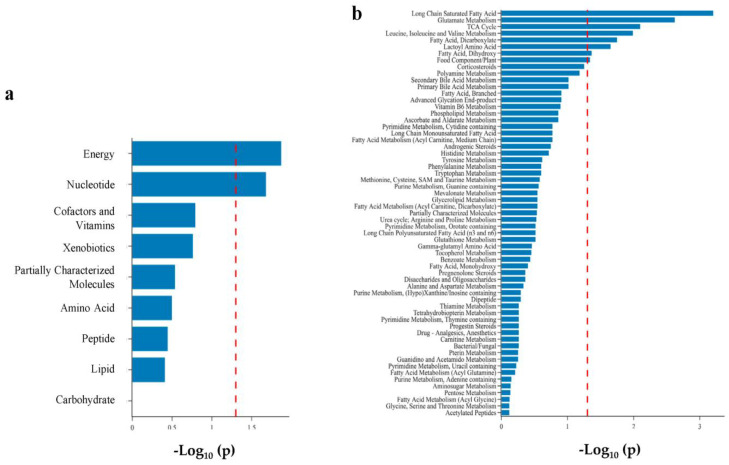
Bar plot of pathway enrichment analysis of metabolomic differences between infants on TPN vs. enteral feeds. (**a**) superpathway; (**b**) subpathway. Metabolite set enrichment analysis was performed using a Fisher Exact Test on subpathways with at least two named metabolites. Eight sub-pathways were found to be enriched at *p* < 0.05 (dashed red line).

**Figure 5 metabolites-13-00971-f005:**
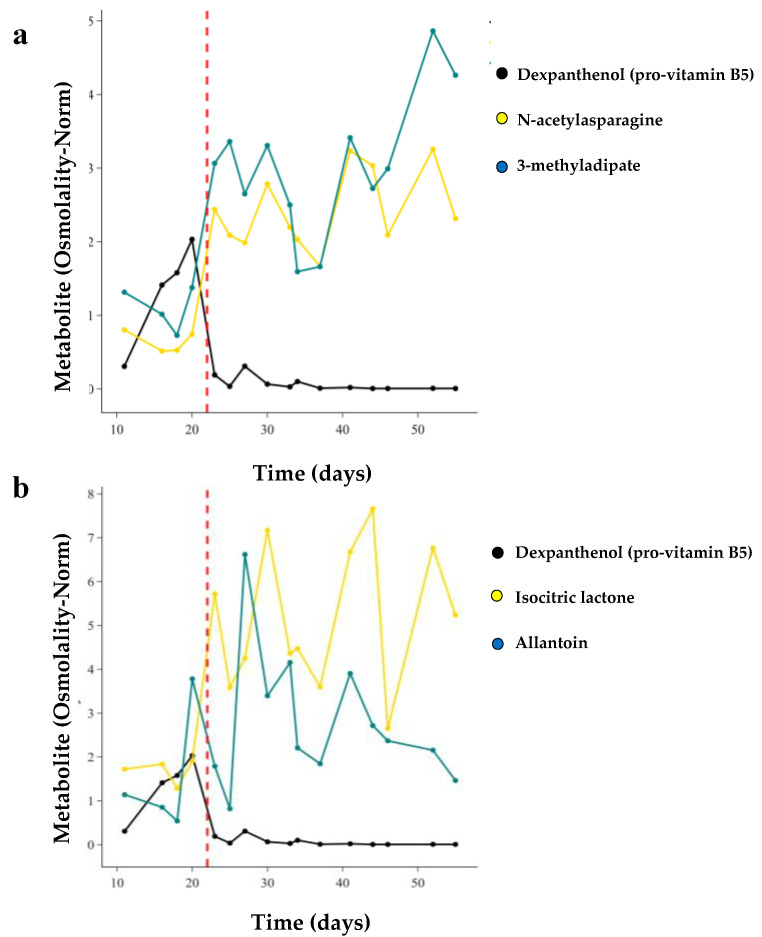
Longitudinal trajectories of individual metabolite concentrations from 9 superpathways for a single infant transitioning off TPN. Dexpanthenol (pro-vitamin B5) was used as a marker of TPN treatment. Dashed red line indicates when the infant was fully off TPN. These trajectories encompasses the top amino acid and lipid metabolites identified in our MWAS (**a**), along with energy and nucleotide pathways (**b**), carbohydrate and cofactor/vitamin pathways (**c**), and peptide and partially characterized molecule super pathways (**d**).

**Table 1 metabolites-13-00971-t001:** Characteristics of infants included in metabolomic studies from TOLSURF and PROP by feeding status at 28 days. Quantitative measurements were compared using a Student’s *t*-test and categorical data using a Fisher’s Exact Test.

	TOLSURF			PROP		
	TPN	Enteral	*p*-Value	TPN	Enteral	*p*-Value
Number of Infants	108	63	-	80	63	-
Age at Sample in days, mean (SD)	27.3 (2.2)	28.1 (2.0)	0.013	28.3 (0.9)	28.0 (0.9)	0.14
Number of Infants by Maternal Race (NHW|AA|HL) *	50|40|18	35|21|7	0.43	35|30|15	30|25|8	0.62
Gestational Age in weeks, mean (SD)	25.2 (1.2)	25.6 (1.2)	0.034	25.7 (1.0)	26.0 (1.3)	0.11
Male sex (%)	58.3	58.7	1.00	51.3	54.0	0.87
Birthweight (g), mean (SD)	698 (164)	778 (181)	0.008	770 (161)	804 (131)	0.18
On corticosteroids at day 28 (%)	22.2	14.3	0.23	28.7	15.9	0.0029

* NHW = Non-Hispanic White; AA = African American; HL = Hispanic Latino.

**Table 2 metabolites-13-00971-t002:** Top 20 most significant metabolites that varied by feeding status at 28 days. Metabolites with OR < 1 are enriched in the urine of infants who had transitioned to enteral feeds while those with OR > 1 are enriched for infants on TPN. AVG FC is the average fold change for enteral vs. TPN feeds from the TOLSURF and PROP cohort.

Biochemical Name	Super-Pathway	Meta *p*	Meta OR	AVG FC
Dexpanthenol	Xenobiotics	6.6 × 10^−15^	11.19	6.30
Gluconate	Xenobiotics	6.7 × 10^−14^	12.06	4.83
Ferulic acid 4-sulfate	Xenobiotics	5.9 × 10^−13^	0.49	0.23
2,6-dihydroxybenzoic acid	Xenobiotics	3.1 × 10^−12^	0.72	0.18
Ascorbic acid 3-sulfate	Cofactors and Vitamins	3.5 × 10^−12^	0.43	0.36
3-methyladipate	Lipid	6.4 × 10^−12^	0.35	0.50
N-acetylasparagine	Amino Acid	2.6 × 10^−11^	0.41	0.43
Methylsuccinate	Amino Acid	3.3 × 10^−11^	0.40	0.35
Acetylhydroquinone sulfate	Xenobiotics	4.2 × 10^−11^	0.28	0.29
N-succinyl-phenylalanine	Amino Acid	4.3 × 10^−11^	0.47	0.28
Lyxonate	Carbohydrate	4.5 × 10^−11^	0.32	0.43
Arabonate/xylonate	Carbohydrate	5.7 × 10^−11^	0.22	0.50
Isocitric lactone	Energy	6.6 × 10^−11^	0.61	0.36
Threonate	Cofactors and Vitamins	7.2 × 10^−11^	0.18	0.61
N-acetyltyrosine	Amino Acid	1.3 × 10^−10^	5.11	7.08
Tartronate (hydroxymalonate)	Xenobiotics	1.7 × 10^−10^	0.36	0.40
3-hydroxy-2-methylpyridine sulfate	Xenobiotics	2.8 × 10^−10^	0.61	0.25
N-delta-acetylornithine	Amino Acid	3.3 × 10^−10^	0.56	0.26
N-acetylphenylalanine	Amino Acid	3.7 × 10^−10^	3.41	4.02
Pantothenate	Cofactors and Vitamins	3.7 × 10^−10^	0.75	0.34

## Data Availability

Due to data confidentiality, the datasets generated and analyzed during the current study are not publicly available and only available for reasonable request.

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
