# Peer review of "The Urinary Metabolomic Fingerprint in Extremely Preterm Infants on Total Parenteral Nutrition vs. Enteral Feeds"

_metabolites, 2023, doi:10.3390/metabo13090971_

Round 1

Reviewer 1 Report

The authors present a comprehensive metabolomics study abour  Total Parenteral Nutrition (TPN) on infants. The idea is good, and the structure is well-organized. 

I have a couple of minor concerns for the manuscript.

For Figure 1 b, generally, the PC1/PC2 value for each sample is only mathematically transformed value. I doubt whether it could be used to compare the difference between these two groups.

Minor spell issues should be checked. For instance, line 81, “<29 wks were” should be “<29 weeks were”.

NA

Author Response

The authors present a comprehensive metabolomics study about Total Parenteral Nutrition (TPN) on infants. The idea is good, and the structure is well-organized. 

I have a couple of minor concerns for the manuscript.

  • For Figure 1 b, generally, the PC1/PC2 value for each sample is only mathematically transformed value. I doubt whether it could be used to compare the difference between these two groups.

Response 1: We express our gratitude to the reviewer for their insightful comment. In our study, Principal Component Analysis (PCA) serves as a dimensionality reduction technique applied to metabolomic data, effectively reducing it to two variables while maximizing information retention. As a linear method, PCA is adept at capturing the overall structure within the dataset. Employing PCA to compare variations in infants' feeding status allows us to effectively illustrate global metabolomic differences. Although our analysis yields a robust p-value indicating these disparities, it's important to clarify that we refrain from singling out specific metabolites responsible for this shift. Instead, our focus in this analysis rests on presenting the collective impact of all metabolites, underscoring a pronounced response. Subsequently, our secondary Metabolome-Wide Association Study (MWAS) delves into individual metabolites that exhibit changes in response to infants' feeding status.

  • Minor spell issues should be checked. For instance, line 81, “<29 wks were” should be “<29 weeks were”.

Response 2: English revisions were edited in the manuscript. Thank you for highlighting this grammatical error.

Reviewer 2 Report

The manuscript reports an urinary untargeted metabolomics study of extremely preterm infants submitted to parenteral nutrition in comparison with enteral nutrition (mainly breast milk). In my opinion, the document presents a relevant topic that will be of interest for the readers of Metabolites, with a well-written content. I have only minor comments/suggestions for the authors:

1) The text is missing information about how the urine samples were collected (collection bag, catheter) and prepared prior to analysis (dilution, centrifugation, etc.).

2) Pg. 3, line 122: It probably should be Fig. S1, instead of Fig. S2.

3) Fig. 4: On the x-axis of both (a) and (b), it should be -log10(p), instead of only log10(p).

4) Fig. 5: Describe what the plots a-d represent (different infants?).

5) Fig. S3: On the x- and y-axis of the (b) plot, specify that -log10 is -log10(p).

Author Response

The manuscript reports an urinary untargeted metabolomics study of extremely preterm infants submitted to parenteral nutrition in comparison with enteral nutrition (mainly breast milk). In my opinion, the document presents a relevant topic that will be of interest for the readers of Metabolites, with a well-written content. I have only minor comments/suggestions for the authors:

  • The text is missing information about how the urine samples were collected (collection bag, catheter) and prepared prior to analysis (dilution, centrifugation, etc.).

Response 1: More information on how the urine samples were collected and processed was added to Section 2.2 Metabolomic Profiling (lines 98-101).

 Urine samples were collected in cotton balls in the diapers, and urine was expressed into collection tubes using a syringe. Samples were frozen at -70°C and shipped on dry ice to Metabolon for untargeted metabolomic profiling.

2) Pg. 3, line 122: It probably should be Fig. S1, instead of Fig. S2.

Response 2: Thank you for your observation. Please notice line 127 to see the order of Figure S2 changed to S1.

3) Fig. 4: On the x-axis of both (a) and (b), it should be -log10(p), instead of only log10(p).

Response 3: Thank you for your observation, Fig 4 x-axis has been changed to -log10(p).

4) Fig. 5: Describe what the plots a-d represent (different infants?).

Response 4: Figure 5 a-d represents the longitudinal metabolomic trajectorey of a single infant across 9 different metabolites, we clarified this detail in the legend of Fig 5. Lines 277-280.

Figure 5. Trajectories of individual metabolite concentrations from 9 super-pathways for a single infant transitioning off TPN. The longitudinal trajectory of all the metabolites shown are for the same infant. Dexpanthenol (pro-vitamin B5) was used as a marker of TPN treatment. Dashed red line indicates when the infant was fully off TPN.

5) Fig. S3: On the x- and y-axis of the (b) plot, specify that -log10 is -log10(p).

Response 5: Thank you for your observation, Fig S3 x- and -y axis has been changed to -log10(p). This can be found at https://zenodo.org/record/8191070.

Reviewer 3 Report

Guardado M et al describes the Urinary Metabolomic Fingerprint in Extremely Preterm In-2 fants on Total Parenteral Nutrition vs Enteral Feeds. They performed untargeted global metabolomics on urine samples collected between 23-30 18 days of life from 314 infants born <29 weeks gestational age from the TOLSURF and PROP cohorts. although the study is very well described I would try to identify, as well as done for NEC other clinical implications of the difference between NP and enteral. They are described in the literature on the role of aminocide and lipid mixtures in the modification of urinary metabolites Therefore I would try to reinforce the clinical significance of this alteration or to limit myself to the description of the alterations without trying to give a possible meaning. It is absolutely predictable that artificial nutrition is profoundly different from enteral. I suggest that I value the study of metabolomics rather than trying to give meaning to differences that are obvious.

Author Response

Guardado M et al describes the Urinary Metabolomic Fingerprint in Extremely Preterm Infants on Total Parenteral Nutrition vs Enteral Feeds. They performed untargeted global metabolomics on urine samples collected between 23-30 18 days of life from 314 infants born <29 weeks gestational age from the TOLSURF and PROP cohorts. although the study is very well described I would try to identify, as well as done for NEC other clinical implications of the difference between NP and enteral. They are described in the literature on the role of aminocide and lipid mixtures in the modification of urinary metabolites Therefore I would try to reinforce the clinical significance of this alteration or to limit myself to the description of the alterations without trying to give a possible meaning. It is absolutely predictable that artificial nutrition is profoundly different from enteral. I suggest that I value the study of metabolomics rather than trying to give meaning to differences that are obvious.

Response 1: We concur with the notion that artificial nutrition distinctly differs from enteral feeds and one of the distinctive features of our study is its exploration by untargeted metabolomics. While we acknowledge the potential confounding effect of premature infants transitioning to enteral feeds and manifesting a metabolomic response, it is noteworthy that numerous vital metabolites essential for infant growth (such as Essential Amino Acids and Lipids) are significantly enriched in infants receiving enteral feeds. This highlights how molecular research aligns with established clinical knowledge. Moreover, we have refined Section 5 (Conclusions) to accentuate the novel metabolomic contributions and clinical implications of our study.

This study demonstrates how metabolomics can potentially contribute to precision medicine by providing data on levels of specific metabolites to allow adjustments in the parenteral nutrition solution, showcasing a molecular investigation in harmony with established clinical understanding.

Reviewer 4 Report

The study is well introduced and the results of the metabolomics analysis are well presented. However, patient data should be implemented, if there were exclusion criteria, associated pathologies before and at the time of urine collection. Also, I think it would be important to know the difference in weight gain between patients on enteral nutrition and those on parenteral nutrition. It should also be clarified whether the human milk was fortified.

Author Response

The study is well introduced and the results of the metabolomics analysis are well presented. However, patient data should be implemented, if there were exclusion criteria, associated pathologies before and at the time of urine collection.

Response 1: Exclusion Criteria can be found at: https://classic.clinicaltrials.gov/ct2/show/NCT01022580 for TOLSURF and https://classic.clinicaltrials.gov/ct2/show/NCT01435187 for PROP.

Also, I think it would be important to know the difference in weight gain between patients on enteral nutrition and those on parenteral nutrition. It should also be clarified whether the human milk was fortified.

Response 2: Thank you for your observation. We acknowledge the absence of available longitudinal data for infant weight. This limitation is addressed and noted as a potential avenue for future investigation (Lines 387-390).

 Furthermore, we could not evaluate relationships between metabolite levels and feeding status with infant weight gain, as longitudinal weight data were not collected. Subsequent research should aim to investigate this aspect.